# Fas cell surface death receptor controls hepatic lipid metabolism by regulating mitochondrial function

Flurin Item[1,2], Stephan Wueest[1,2], Vera Lemos[3], Sokrates Stein[3], Fabrizio C. Lucchini[1,2,4], Rémy Denzler[5,6], Muriel C. Fisser[5,6], Tenagne D. Challa[1,2], Eija Pirinen[7,8], Youngsoo Kim[9], Silvio Hemmi[10], Erich Gulbins[11], Atan Gross[12], Lorraine A. O'Reilly[13,14], Markus Stoffel[5,6], Johan Auwerx[7] & Daniel Konrad[1,2,4]

Nonalcoholic fatty liver disease is one of the most prevalent metabolic disorders and it tightly associates with obesity, type 2 diabetes, and cardiovascular disease. Reduced mitochondrial lipid oxidation contributes to hepatic fatty acid accumulation. Here, we show that the Fas cell surface death receptor (Fas/CD95/Apo-1) regulates hepatic mitochondrial metabolism. Hepatic Fas overexpression in chow-fed mice compromises fatty acid oxidation, mitochondrial respiration, and the abundance of mitochondrial respiratory complexes promoting hepatic lipid accumulation and insulin resistance. In line, hepatocyte-specific ablation of Fas improves mitochondrial function and ameliorates high-fat-diet-induced hepatic steatosis, glucose tolerance, and insulin resistance. Mechanistically, Fas impairs fatty acid oxidation via the BH3 interacting-domain death agonist (BID). Mice with genetic or pharmacological inhibition of BID are protected from Fas-mediated impairment of mitochondrial oxidation and hepatic steatosis. We suggest Fas as a potential novel therapeutic target to treat obesity-associated fatty liver and insulin resistance.

[1] Division of Pediatric Endocrinology and Diabetology, University Children's Hospital, CH-8032 Zurich, Switzerland. [2] Children's Research Center, University Children's Hospital, CH-8032 Zurich, Switzerland. [3] Metabolic Signaling, École Polytechnique Fédérale de Lausanne (EPFL), CH-1015 Lausanne, Switzerland. [4] Zurich Center for Integrative Human Physiology, University of Zurich, CH-8057 Zurich, Switzerland. [5] Institute of Molecular Health Sciences, ETH Zurich, CH-8093 Zurich, Switzerland. [6] Competence Center of Systems Physiology and Metabolic Disease, ETH Zurich, CH-8093 Zurich, Switzerland. [7] Laboratory of Integrative and Systems Physiology (LISP), École Polytechnique Fédérale de Lausanne (EPFL), CH-1015 Lausanne, Switzerland. [8] Department of Biotechnology and Molecular Medicine, A.I. Virtanen Institute for Molecular Sciences, University of Eastern Finland, P.O. Box 1627, FIN-70211 Kuopio, Finland. [9] Ionis Pharmaceuticals Inc., Carlsbad, 92010 California, USA. [10] Institute of Molecular Life Sciences, University of Zurich, CH-8057 Zurich, Switzerland. [11] Department of Molecular Biology, University of Duisburg-Essen, Essen, D-45147, Germany. [12] Department of Biological Regulation, The Weizmann Institute of Science, Rehovot 76100, Israel. [13] The Walter and Eliza Hall Institute of Medical Research, Parkville, Victoria 3052, Australia. [14] Department of Medical Biology, The University of Melbourne, Parkville, Victoria 3050, Australia. Correspondence and requests for materials should be addressed to D.K. (email: daniel.konrad@kispi.uzh.ch)

Aberrant accumulation of lipids in the liver correlates with abdominal obesity and insulin resistance[1]. It constitutes a hallmark of nonalcoholic fatty liver disease (NAFLD), which has emerged as the most common chronic liver disease in the industrialized world affecting ~30% of the adult population and an increasing number of children[2]. The spectrum of NAFLD ranges from simple nonprogressive fatty liver (hepatic steatosis) to the potentially progressive nonalcoholic steatohepatitis, which can proceed to fibrosis, cirrhosis, and hepatocellular carcinoma[3]. As NAFLD is becoming increasingly common in countries with predominantly sedentary life styles, it is predicted to become the leading indication for liver transplantation in the United States within the next 5 years[4]. However, the molecular mechanisms underlying the pathogenesis of obesity-associated fatty liver disease is poorly understood.

Fas (CD95/Apo-1) is a cell surface glycoprotein belonging to the tumor necrosis factor (TNF) receptor superfamily and is constitutively expressed in most tissues. Ligation of the Fas receptor initiates proteolytic cleavage of intracellular caspases culminating in apoptosis in many cell types. In addition to its well-established role in apoptosis, activation of Fas can also induce diverse nonapoptotic signaling pathways depending on the tissue and conditions[5]. Evidence from animal and human studies indicates that Fas and/or its ligand FasL are upregulated in fatty liver disease[6–8]. Moreover, hepatocytes in fatty livers are hypersensitive to Fas-mediated apoptosis causing hepatic injury that eventually leads to cirrhosis and end-stage liver disease[8]. Conversely, Fas antagonism tempers hepatocyte apoptosis and liver damage. These studies indicated that Fas may mediate increased susceptibility of NAFLD to end-stage liver disease such as cirrhosis and liver carcinoma[8]. However, it is not known whether Fas contributes to the development of hepatic lipid accumulation. Furthermore, the metabolic consequence of Fas activation in the liver has not yet been determined. In order to assess a potential role of hepatic Fas in obesity-induced metabolic dysregulation, mice with liver-specific Fas depletion or over-expression were generated. Herein, we demonstrate that Fas regulates hepatic mitochondrial function and fatty acid oxidation and, hence, contributes to the development of hepatic steatosis. Furthermore, pharmacological Fas depletion in the liver of obese mice protects them from the development of hepatic steatosis and insulin resistance, rendering Fas a promising novel therapeutic target.

## Results

**Fas knockout in the liver reduces diet-induced steatosis.** To test a putative role of hepatic Fas activation in obesity-induced metabolic dysregulation, liver-specific Fas-knockout mice (Fas$^{flox/flox}$, Alb-Cre$^{+/−}$; Fas$^{\Delta hep}$) were generated using the cre-lox system[9]. As a control, littermate mice with floxed Fas but absent Cre-recombinase (Cre) expression were used (Fas$^{flox/flox}$, Alb-Cre$^{−/−}$; Fas$^{F/F}$). Western blot analysis confirmed depletion of Fas protein specifically in the liver of Fas$^{\Delta hep}$ mice whereas it was unchanged in all other tissues analyzed (Supplementary Fig. 1a). In order to investigate the physiological significance of hepatocyte-specific Fas depletion, mice were fed either standard chow or high-fat diet (HFD) for 6 weeks. Total body weight gain was similar in Fas$^{\Delta hep}$ and their Cre-negative littermates under each diet (Fig. 1a). Similarly, no differences were observed in fat pad weights (Supplementary Fig. 1b). Metabolic phenotyping revealed comparable food intake, metabolic rate, respiratory quotient, and locomotor activity in both genotypes (Supplementary Fig. 1c). Importantly, accumulation of hepatic lipid droplets was reduced in HFD-fed Fas$^{\Delta hep}$ mice as assessed by histological analysis (Fig. 1b). Consistently, hepatic triglyceride

(TG) content was significantly lower in HFD-fed Fas$^{\Delta hep}$ compared to Fas$^{F/F}$ mice (Fig. 1c). Intrahepatic accumulation of TGs generates various deleterious lipid intermediates, including ceramide and diacylglycerols (DAGs). Both lipids may interfere with canonical insulin signaling, thus possibly affecting insulin sensitivity[10, 11]. Notably, ceramide and DAG levels were significantly reduced in the liver of HFD-fed Fas$^{\Delta hep}$ compared to control mice (Fig. 1c). In contrast, neither plasma TG nor free-fatty acid (FFA) levels differed between both groups (Table 1). Similarly, circulating insulin, adiponectin, and leptin concentrations were unchanged (Table 1). Of note, similar serum concentrations of alanine aminotransferase (ALT) and aspartate aminotransferase (AST) (Supplementary Fig. 1d) as well as similar hepatic protein levels of cleaved caspase 3 and poly (ADP-ribose) polymerase (PARP) (Supplementary Fig. 1e) were detected, suggesting a similar degree of activation of the apoptotic pathway in both groups of HFD-fed mice. In addition, plasma and hepatic proinflammatory profile was not significantly altered in HFD-fed Fas$^{\Delta hep}$ mice (Supplementary Fig. 1f, g).

The storage of excess lipids in insulin-sensitive organs is strongly associated with deteriorated glucose tolerance and insulin resistance[11]. HFD feeding for 6 weeks impaired glucose tolerance in Fas$^{F/F}$ compared to chow-fed mice. Strikingly, Fas$^{\Delta hep}$ mice were partly protected from deteriorated glucose metabolism (Fig. 1d). Lower gluconeogenic conversion of pyruvate to glucose as assessed by the pyruvate tolerance test further confirmed improved glucose homeostasis in HFD-fed Fas$^{\Delta hep}$ mice (Fig. 1e). To assess insulin sensitivity in HFD-fed mice, hyperinsulinemic-euglycemic clamp studies were performed. A significantly increased glucose infusion rate in HFD-fed Fas$^{\Delta hep}$ compared to Fas$^{F/F}$ mice was noted, consistent with improved whole-body insulin sensitivity (Fig. 1f and Supplementary Fig. 1h, i). Importantly, endogenous glucose production under hyperinsulinemic clamp conditions was significantly reduced in the absence of hepatic Fas compared to control littermates indicating improved hepatic insulin sensitivity (Fig. 1f). In contrast, the insulin-stimulated glucose disposal rate did not differ between either genotype, suggesting similar muscle insulin sensitivity (Supplementary Fig. 1j, k). Corroborating this finding, insulin-stimulated Akt (Thr308) phosphorylation was elevated in liver lysates from HFD-fed Fas$^{\Delta hep}$ compared to Fas$^{F/F}$ mice (Fig. 1g and Supplementary Fig. 1l). Collectively, these results indicate that hepatic Fas depletion protects mice at least partly from obesity-induced lipid accumulation and insulin resistance in the liver.

**Fas overexpression leads to steatosis and insulin resistance.** To examine whether hepatic Fas expression is sufficient to induce liver lipid accumulation and insulin resistance, C57BL/6J mice were injected with an adenoviral vector expressing Fas (Ad-Fas). Fas protein levels were ~1.8-fold higher in livers of mice receiving Ad-Fas compared to mice receiving a control vector expressing lacZ (Ad-lacZ) (Fig. 2a). At 12 days after injection, body weight was similar in both groups of mice (Fig. 2b), whereas blood glucose but not plasma insulin, TG, and FFA levels were increased in Fas-overexpressing mice (Supplementary Fig. 2a). Hepatic Fas overexpression resulted in hepatic steatosis as demonstrated by a significant higher TG content (Fig. 2c). Moreover, lipid intermediates such as ceramide and DAG accumulated in livers of Ad-Fas mice (Fig. 2c). An intraperitoneal glucose tolerance test was impaired (Fig. 2d) and hyperinsulinemic-euglycemic clamp studies revealed significantly decreased glucose infusion rate and, importantly, higher endogenous glucose production under clamp conditions in Ad-Fas-injected vs. control animals (Fig. 2e and Supplementary

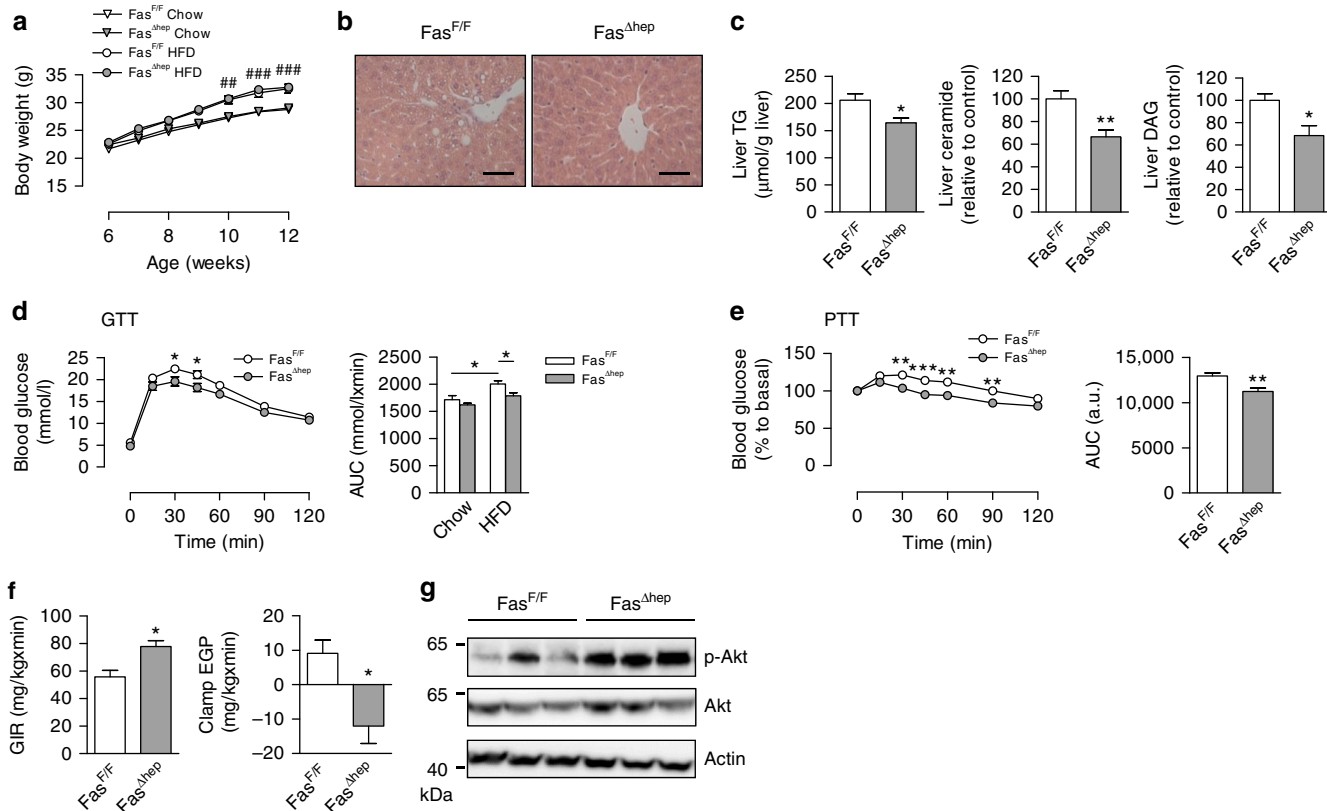

**Fig. 1** Conditional hepatic Fas knockout ameliorates diet-induced hepatic steatosis and insulin resistance. **a** Body weight gain during 6 weeks of chow (Fas$^{F/F}$, $n=18$; Fas$^{\Delta hep}$, $n=17$) or HFD feeding (Fas$^{F/F}$, $n=22$; Fas$^{\Delta hep}$, $n=24$). **b** Hematoxylin and eosin (H&E)-stained liver sections from HFD-fed Fas$^{F/F}$ and Fas$^{\Delta hep}$ mice. *Scale bar* represents 100 μm. **c** Liver TG (Fas$^{F/F}$, $n=8$; Fas$^{\Delta hep}$, $n=7$), ceramide, and DAG (Fas$^{F/F}$, $n=5$; Fas$^{\Delta hep}$, $n=6$) levels are shown. **d** Intraperitoneal glucose tolerance test (chow-fed: Fas$^{F/F}$, $n=5$; Fas$^{\Delta hep}$, $n=6$; HFD-fed: Fas$^{F/F}$, $n=12$; Fas$^{\Delta hep}$, $n=9$) and **e** intraperitoneal pyruvate tolerance test in HFD-fed mice (Fas$^{F/F}$, $n=9$; Fas$^{\Delta hep}$, $n=10$) at 12 weeks of age. **f** Glucose infusion rate (GIR) and endogenous glucose production (EGP) during hyperinsulinemic-euglycemic clamps, $n=5$. **g** Representative western blots of total liver lysate of HFD-fed Fas$^{F/F}$ and Fas$^{\Delta hep}$ mice. Values are expressed as mean±s.e.m.; *$p < 0.05$, **$p < 0.01$, and ***$p < 0.001$ indicate significant differences between genotypes and ##$p < 0.01$ and ###$p < 0.001$ between diets. Statistical tests used: *t*-tests for (**c**, **e** (*right panel*), **f**); ANOVA for (**a**, **d**, **e** (*left panel*)). AUC area under the curve

**Table 1 Phenotypic characteristics of chow- and HFD-fed Fas$^{F/F}$ and Fas$^{\Delta hep}$ mice**

|  | Fas$^{F/F}$ chow | Fas$^{\Delta hep}$ chow | Fas$^{F/F}$ HFD | Fas$^{\Delta hep}$ HFD |
|---|---|---|---|---|
| Glucose (mmol/l) | $9.2 \pm 0.5$ ($n=5$) | $7.7 \pm 0.4$ ($n=6$) | $10.6 \pm 0.5$ ($n=9$) | $10.3 \pm 0.5$## ($n=8$) |
| Insulin (pmol/l) | $165 \pm 26$ ($n=5$) | $145 \pm 21$ ($n=5$) | $332 \pm 29$## ($n=9$) | $264 \pm 31$ ($n=6$) |
| FFA (mmol/l) | $0.57 \pm 0.06$ ($n=5$) | $0.64 \pm 0.13$ ($n=6$) | $0.82 \pm 0.06$ ($n=9$) | $0.77 \pm 0.07$ ($n=8$) |
| TG (mg/dl) | $66.2 \pm 5.9$ ($n=5$) | $60.9 \pm 4.3$ ($n=6$) | $86.5 \pm 6.2$ ($n=9$) | $82.8 \pm 6.8$ ($n=8$) |
| Adiponectin (μg/ml) | $7.2 \pm 0.7$ ($n=5$) | $7.1 \pm 0.3$ ($n=6$) | $6.2 \pm 0.4$ ($n=9$) | $5.7 \pm 0.3$ ($n=8$) |
| Leptin (pg/ml) | $131 \pm 44$ ($n=5$) | $65 \pm 14$ ($n=6$) | $436 \pm 87$# ($n=9$) | $338 \pm 37$# ($n=8$) |

Mice were fasted for 5 h before blood sampling. Values are expressed as mean±s.e.m.
#$P < 0.05$ and ##$p < 0.01$ indicate significant differences between diets of the same genotype (ANOVA)

Fig. 2b–d). In line with impaired hepatic insulin sensitivity, insulin-stimulated hepatic Akt (Thr308) phosphorylation was significantly diminished in Ad-Fas compared to control Ad-lacZ mice (Fig. 2f and Supplementary Fig. 2e). Taken together, hepatic Fas overexpression was sufficient to induce steatosis and impaired insulin sensitivity, suggesting that Fas is a potent regulator of both hepatic lipid metabolism and insulin sensitivity.

**Fas regulates mitochondrial function**. Hepatic steatosis may be the result of reduced lipid oxidation[12–14]. In order to determine whether Fas activation may impair mitochondrial fatty acid oxidation, we measured oleic acid oxidation in liver homogenates of

Fas-overexpressing mice. Oleic acid oxidation rates were significantly decreased by ∼22% in Ad-Fas-treated mice (Fig. 3a). However, mitochondrial DNA abundance was not changed (Supplementary Fig. 3a), suggesting that Fas-mediated reduction in hepatic mitochondrial function is not due to decreased mitochondrial number. Of note, Ad-Fas injection did neither induce cytochrome *c* release nor increase cleavage of caspase 3 in the liver (Supplementary Fig. 3b, c). In agreement with such notion, protein levels of the large fragment of PARP, which result from caspase cleavage and is involved in DNA damage detection and repair, were comparable between Ad-Fas and Ad-lacZ mice (Supplementary Fig. 3d), suggesting that impaired mitochondrial function is not the result of induced apoptosis.

Besides a reduction in lipid oxidation, elevated de novo lipogenesis, increased fatty acid uptake, or blunted TG secretion may contribute to hepatic steatosis[1, 15]. In hepatic Fas-overexpressing mice, protein levels of the active form of the lipogenic transcription factor sterol regulatory element-binding protein 1 (SREBP1) was slightly lower (Supplementary Fig. 3e). In addition, hepatic mRNA expression of key genes involved in lipogenesis and fatty acid transport were unchanged or reduced in Ad-Fas compared to Ad-LacZ mice (Supplementary Fig. 3f), suggesting that neither de novo lipogenesis nor fatty acid uptake was significantly affected by hepatic Fas overexpression. However, hepatic triglycerides secretion as measured by Triton-induced hypertriglyceridemia was significantly reduced in Ad-Fas compared to Ad-LacZ mice (Supplementary Fig. 3g). Similarly,

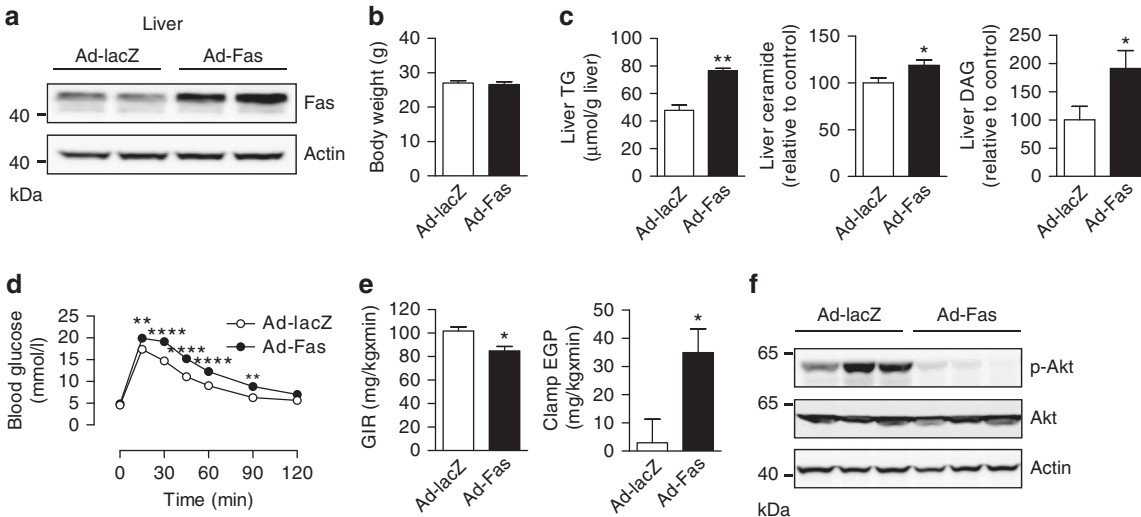

**Fig. 2** Adenovirus-mediated hepatic overexpression of Fas induces hepatic steatosis and hepatic insulin resistance. **a** Protein levels of Fas in total liver lysate of mice injected with adenoviruses expressing either Fas (Ad-Fas) or a control vector (Ad-lacZ). **b** Body weight of mice 12 days after adenovirus injection, $n = 7$. **c** Liver TG ($n = 5$), ceramide, and DAG (Ad-LacZ, $n = 6$; Ad-Fas, $n = 7$) levels are shown. **d** Intraperitoneal glucose tolerance test ($n = 7$), **e** glucose infusion rate (GIR), and endogenous glucose production (EGP) during hyperinsulinemic-euglycemic clamps ($n = 4$) in mice 12 days after injection of Ad-lacZ or Ad-Fas are depicted. **f** Representative western blots of total liver lysate harvested from mice 15–16 days after injection of Ad-lacZ or Ad-Fas. Values are expressed as mean±s.e.m.; *$p < 0.05$, **$p < 0.01$, and ****$p < 0.0001$. Statistical tests used: $t$-test for (**b**, **c** (TG, ceramide), **e**); Mann–Whitney for (**c** (DAG)); ANOVA for (**d**)

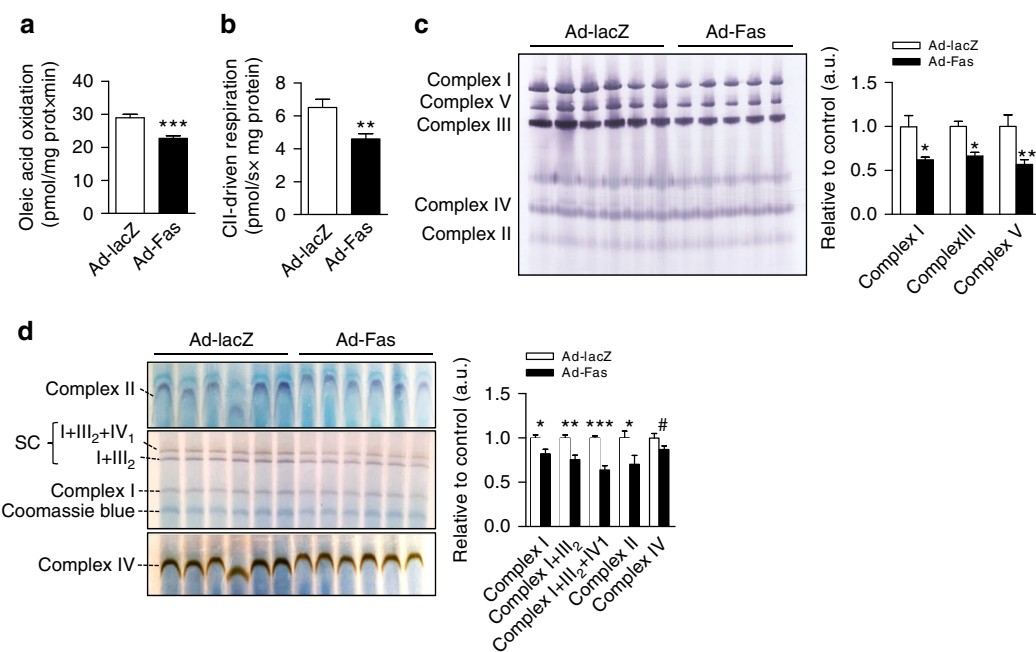

**Fig. 3** Fas impacts on hepatic mitochondrial function and fatty acid oxidation. **a** Oleic acid oxidation rate in liver homogenates of mice injected with adenovirus expressing Fas (Ad-Fas) or lacZ (Ad-lacZ), $n = 7$. **b** Respirometry analysis of complex II (CII)-driven respiration in liver tissue from Ad-Fas ($n = 11$) and Ad-lacZ ($n = 16$) mice. **c** Blue native (BN) polyacrylamide gel electrophoresis (PAGE) and **d** in-gel activity assay using isolated mitochondria from liver of Ad-Fas ($n = 5$ and 6 respectively) and Ad-lacZ ($n = 6$) mice. Quantification of individual band was performed using ImageJ. Values are expressed as mean±s.e.m.; *$p < 0.05$, **$p < 0.01$, ***$p < 0.001$, and #$p = 0.08$ (Student's $t$-test)

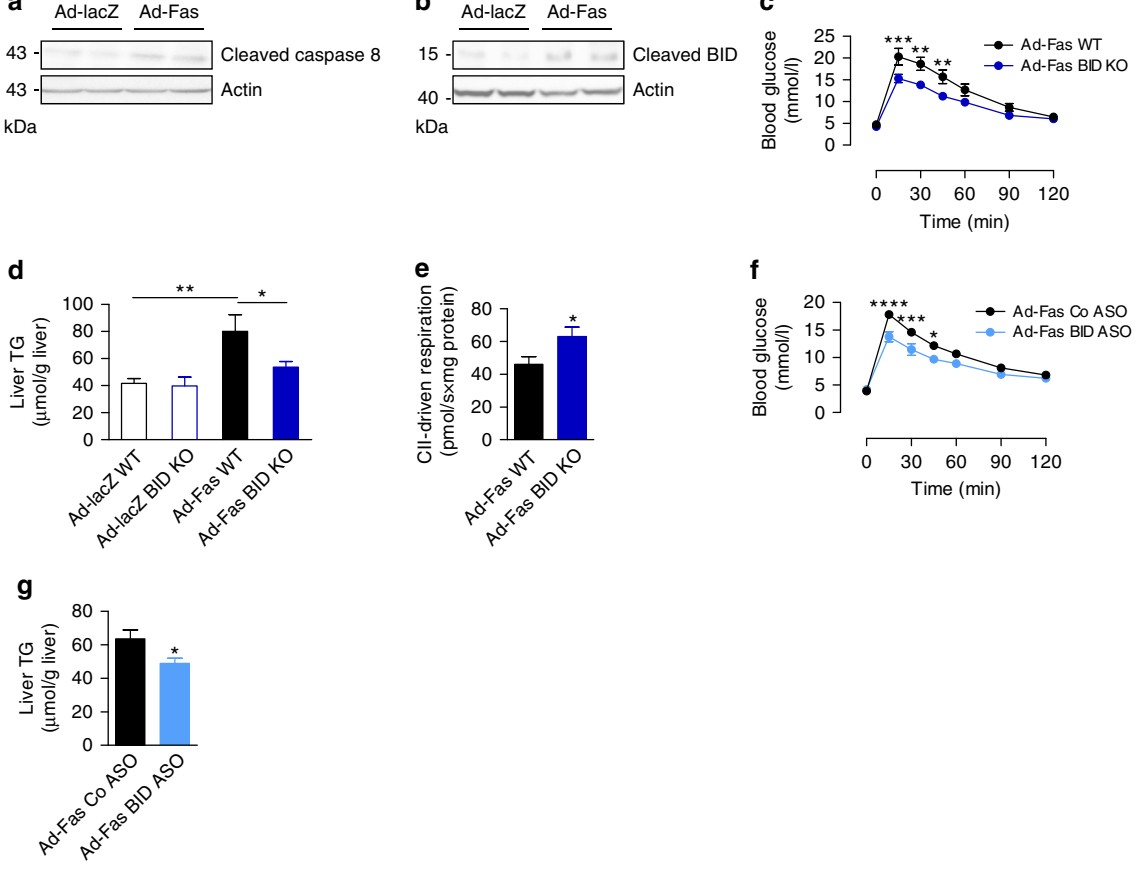

**Fig. 4** Fas impairs mitochondrial function BID-dependently. Protein levels of **a** cleaved caspase 8 (p43) and **b** cleaved BID in total liver lysate harvested from mice 15–16 days after injection of Ad-lacZ or Ad-Fas. **c** Intraperitoneal glucose tolerance test (Ad-Fas WT, $n = 4$; Ad-Fas BID KO, $n = 6$), **d** liver TG levels (Ad-lacZ WT, $n = 4$; Ad-lacZ BID KO, $n = 5$; Ad-Fas WT, $n = 4$; Ad-Fas BID KO, $n = 6$) and **e** hepatic mitochondrial respiration in BID-knockout (*blue*, $n = 10$) and WT (*black*, $n = 9$) mice 12 days after injection of Ad-lacZ or Ad-Fas are depicted. **f** Intraperitoneal glucose-tolerance test ($n = 6$) and **g** liver TG levels ($n = 4$) in mice that received antisense oligonucleotides against BID (BID-ASO; *light blue*) or control-ASO (*black*) 12 days after injection Ad-Fas are depicted. Values are expressed as mean±s.e.m.; *$p < 0.05$, **$p < 0.01$, and ***$p < 0.001$. Statistical tests used: *t*-test for (**e**, **g**) and ANOVA for (**c**, **d**, **f**)

steady-state mRNA levels of *microsomal triglyceride transfer protein* (*Mttp*), a key protein involved in triglyceride secretion[16] (Supplementary Fig. 3f), were slightly reduced. In HFD-fed Fas$^{\Delta hep}$ mice, cleaved SREBP1 protein and transcript levels of aforementioned genes including *Mttp* were unchanged when compared to Fas$^{F/F}$ mice (Supplementary Fig. 3h, i). Thus, the development of Fas-mediated hepatic steatosis may mainly result from impaired mitochondrial function.

To further substantiate such hypothesis, oxygen consumption was analyzed in liver tissue from Ad-Fas mice by high-resolution respirometry. Oxygen consumption rate driven by succinate was significantly lower in liver mitochondria of Ad-Fas compared to Ad-lacZ mice, indicating reduced complex II-driven respiration in these mice (Fig. 3b). To elucidate whether Fas compromises the expression levels and activity of the respiratory chain complexes, blue native (BN) polyacrylamide gel electrophoresis (PAGE) analysis and in-gel activity assays of hepatic mitochondrial complexes were performed. Increased hepatic Fas expression significantly reduced the abundance of complexes I, III, V (Fig. 3c). More importantly, the activity of complex II, and complexes I and IV, both individually and in supercomplexes decreased upon Fas overexpression (Fig. 3d). Conversely, Fas ablation in hepatocytes had the opposite effect: the abundance of complex I and, to a lesser extent, of complex III was increased in obese Fas$^{\Delta hep}$ mice compared to Fas$^{F/F}$controls (Supplementary Fig. 3j). Taken together, these results suggest that Fas not only

impairs fatty acid oxidation in hepatocytes but also affects abundance and function of mitochondrial complexes.

**Fas impairs mitochondrial function BID-dependently.** Fas induces caspase 8-mediated cleavage of the BH3 interacting-domain death agonist (BID), thereby triggering mitochondrial outer membrane permeabilization, leading to mitochondrial damage[17, 18]. Hence, Fas activation may induce mitochondrial dysfunction and decrease lipid oxidation in a BID-dependent manner. Indeed, hepatic overexpression of Fas in vivo induced cleavage of caspase 8 as well as BID (Fig. 4a, b and Supplementary Fig. 4a, b). To address a potential role of BID in mediating the effect of Fas on mitochondrial function, Fas was overexpressed in hepatocytes of BID knockout (KO) mice. Of note, BID ablation protected mice from Fas-induced deterioration in glucose tolerance (Fig. 4c), whereas glucose tolerance was similar in wild-type (WT) and BID KO mice upon lacZ injection (Supplementary Fig. 4c). In line with such finding, liver TG levels of Ad-Fas-treated BID KO mice were significantly lower compared to Ad-Fas-treated WT mice and comparable to Ad-lacZ controls (Fig. 4d). Furthermore, mitochondrial complex II-driven respiration was increased in Fas-overexpressing BID KO mice (Fig. 4e). Importantly, treatment with antisense oligonucleotides (ASO) targeting *BID*, which reduced hepatic BID protein content by ~60% (Supplementary Fig. 4d), improved glucose tolerance

and reduced liver TG levels in mice overexpressing Fas in hepatocytes (Fig. 4f, g). These findings strongly support a mechanistic role for BID in mediating the inhibitory effect of Fas on mitochondrial fatty acid oxidation.

In addition, Fas signaling may impact on liver lipid metabolism via activation of acid sphingomyelinase (ASM), which catalyzes the hydrolysis of sphingomyelin to ceramide in various cell types[19, 20]. Therefore, Fas may directly induce the synthesis of toxic lipid metabolites such as ceramide, which in turn may induce hepatic insulin resistance and steatosis. However, ASM activity was not altered in mice with adenovirus-mediated increased hepatic Fas expression (Supplementary Fig. 4e). Accordingly, glucose tolerance as well as liver TG content were similar in hepatic Fas-overexpressing ASM KO mice compared to WT mice (Supplementary Fig. 4f, g), suggesting that ASM activation does not contribute to Fas-mediated changes in hepatic metabolism. In summary, Fas activation reduces mitochondrial function and fatty acid oxidation in hepatocytes in BID-dependent manner.

**Pharmacological depletion of Fas ameliorates steatosis.** To explore the therapeutic potential of targeting hepatic Fas signaling to improve hepatic steatosis and insulin resistance, obese WT mice were treated with ASO against *Fas*[21]. In a first set of experiments, C57BL/6J littermate mice were fed a HFD for 20 weeks and during the last 10 weeks of HFD received either Fas-ASO or eight-base mismatch control oligonucleotide (control-ASO) once a week. Fas-ASO treatment reduced hepatic Fas protein content by ∼90% compared to control-ASO (Supplementary Fig. 5a). Hepatic Fas depletion did not affect body weight, but was associated with lower blood glucose, plasma insulin, FFA, and TG concentrations (Table 2). Importantly, liver TG levels were significantly lower (Fig. 5a) and glucose tolerance significantly improved in HFD-fed Fas-ASO-treated mice (Fig. 5b). Consistent with findings in genetically Fas-depleted mice, pharmacological depletion of hepatic Fas signaling significantly increased mitochondrial respiration (Fig. 5c). Moreover, mitochondrial DNA abundance was increased in Fas-ASO-treated mice (Fig. 5d), suggesting that Fas silencing resulted in increased mitochondrial number. Finally, the expression of mitochondrial complex I and complex V was increased in HFD-fed mice upon pharmacological Fas silencing (Fig. 5e).

In a second set of ASO experiments, leptin-deficient *ob/ob* mice received either Fas-ASO or control-ASO twice a week for a total of 4 weeks. Importantly, Fas-ASO-treated *ob/ob* mice also displayed significantly lower hepatic TG concentrations (Fig. 5f) as well as improved glucose tolerance (Fig. 5g). Moreover, the glucose infusion rate was significantly increased and endogenous glucose production under clamp conditions was lower in Fas-depleted *ob/ob* mice (Fig. 5h and Supplementary Fig. 5b–d) indicating improved hepatic insulin sensitivity. As outlined above,

fatty liver disease is associated with elevated FasL levels. Of note, deficiency of secreted FasL[22] in obese mice (HFD-fed FasL$^{\Delta s/\Delta s}$ mice) reduced liver triglyceride levels (Supplementary Fig. 6). Collectively, these data provide strong evidence that hepatic Fas signaling may be a pharmacological target to treat fatty liver disease and to improve hepatic insulin sensitivity.

## Discussion

Dysregulated hepatic lipid metabolism plays an important role in various metabolic diseases. Particularly, aberrant accumulation of lipids may impair hepatocyte function promoting liver injury and insulin resistance[23, 24]. The present study identifies Fas as a physiological regulator of hepatic mitochondrial function. Furthermore, it suggests that Fas activation in hepatocytes contributes to obesity-associated fatty liver and insulin resistance by impairing mitochondrial fatty acid oxidation.

Mitochondrial fatty acid oxidation is the dominant oxidative pathway for the removal of fatty acids under normal physiological conditions[23], and its impairment may result in the accumulation of liver lipids and, thus, lead to NAFLD[12–14]. Herein, we show that increased hepatic Fas expression compromised mitochondrial fatty acid oxidation, respiration, and the abundance of respiratory complexes. Conversely, depletion of hepatic Fas enhanced mitochondrial respiration and the abundance of respiratory complexes. Changes in TG, ceramide, and DAG levels in the liver paralleled observed alterations in mitochondrial function. Such findings support the notion that impaired mitochondrial fatty acid oxidation leads to accumulation of toxic lipids such as ceramide and DAG[10, 25]. Recent studies have postulated that the latter may interfere with the insulin signaling cascade, thereby causing insulin resistance[10, 11]. Consistently, we present herein that increased hepatic Fas expression induced both ceramide and DAG formation, which was associated with perturbed insulin signaling and increased hepatic insulin resistance. On the other hand, hepatocyte-specific Fas-depleted obese mice had reduced levels of ceramide and DAG in the liver and were protected from the development of HFD-induced hepatic insulin resistance. Interestingly, besides the detrimental effect of ceramide on insulin signaling and, thus, insulin sensitivity, in vitro studies revealed a negative effect of ceramides on mitochondrial respiration[26, 27] that may further promote its own accumulation and, thus, the creation of a vicious cycle. Of note, similar expression levels of genes involved in de novo lipogenesis and fatty acid transport further support a major role of altered mitochondrial function in the observed liver phenotypes. However, we cannot fully exclude that other pathways involved in hepatic lipid metabolism contributed to Fas-mediated hepatic steatosis. In fact, Ad-Fas-injected mice revealed reduced triglyceride secretion, possibly contributing to elevated liver steatosis in these mice.

Fas ligation is well known to induce a cell-intrinsic apoptotic pathway resulting in mitochondrial damage. Activation of Fas induces caspase 8-mediated cleavage of the BH3-only BID protein to the cleaved form cBID, which then translocates to the mitochondria to activate Bax and Bak, resulting in mitochondrial outer membrane permeabilization and cytochrome *c* release[18, 28]. Thus, this mechanism postulates a direct effect of Fas on mitochondrial function and, consequently, accumulation of hepatic lipid metabolites. In accordance with such notion, hepatic overexpression of Fas in vivo induced cleavage of caspase 8 as well as BID and impaired oleic acid oxidation. BID-deficient mice were protected from Fas-induced mitochondrial dysfunction and hepatic steatosis. Consistently, BID-deficient mice were previously found to be protected from HFD-induced hepatic lipid accumulation[29]. Thus, Fas inhibits mitochondrial fatty acid

**Table 2 Phenotypic characteristics of long-term HFD-fed mice treated with antisense oligonucleotides (ASO) against Fas**

|  | Control-ASO | Fas-ASO |
|---|---|---|
| Body weight (g) | 37.4 ± 2.1 | 34.6 ± 2.6 |
| Glucose (mmol/l) | 9.7 ± 0.5 | 8.4 ± 0.4$^{\#}$ |
| Insulin (pmol/l) | 483 ± 78 | 236 ± 42* |
| FFA (mmol/l) | 0.49 ± 0.03 | 0.39 ± 0.03* |
| TG (mg/dl) | 85.5 ± 3.7 | 54.7 ± 4.7*** |

Mice were fasted for 5 h before blood sampling. Values are expressed as mean±s.e.m., n = 5
*$P < 0.05$, ***$p < 0.001$, and $^{\#}p = 0.08$ (Student's *t*-test)

oxidation via BID, thereby contributing to the pathogenesis of obesity-associated hepatic steatosis. Of note, mice with hepatocyte-specific caspase 8 deficiency are protected from the development of methionine-and-choline deficient diet-induced steatosis[30], further supporting a role of a Fas–caspase 8–BID pathway in the pathogenesis of steatosis. Surprisingly, we could neither detect increased cytochrome *c* release nor cleaved caspase 3 in livers of Fas-overexpressing mice. While we cannot exclude that undetected apoptosis may be present in these mice, our data indicate that the degree of Fas-mediated BID cleavage may not be sufficient to trigger cytochrome *c* release and subsequent cleavage of caspase 3. Alternatively, increased protein levels of X-linked inhibitor of apoptosis protein as observed in Ad-Fas-overexpressing mice ($1.0 \pm 0.1$ in Ad-lacZ vs. $1.6 \pm 0.1$ in Ad-Fas, $p < 0.01$) may be responsible for lacking cleavage of caspase 3. Clearly, further studies are required to shed more light on the complex molecular mechanisms involved in the activation of BID-induced apoptotic signaling[31].

ASOs are designed to bind to targeted mRNA by Watson–Crick base pairing, resulting in modulation of its function through a variety of post-binding events[32]. Currently, multiple clinical trials using ASOs are ongoing, including the recent FDA approval of Kynamro™ for homozygous familial hypercholesterolemia[33] and Spinraza® for patients with spinal muscular atrophy[34]. Therefore, ASO treatment has emerged as a promising new approach to treat multiple diseases[32, 35]. Pertinently, it has been reported that mice treated with Fas-ASO exhibited protection against agonistic Fas antibody-induced fulminant hepatitis. Consequently, it was suggested that Fas-ASO may have therapeutic potential in liver disease[21]. We observed herein that pharmacological Fas depletion in the liver of obese mice with Fas-ASOs improved mitochondrial function and concomitantly protected them from the development of hepatic steatosis and insulin resistance. Thus, our results extend the potential use of Fas-ASO treatment to fatty liver disease, i.e., to improve obesity-associated mitochondrial dysfunction, NAFLD,

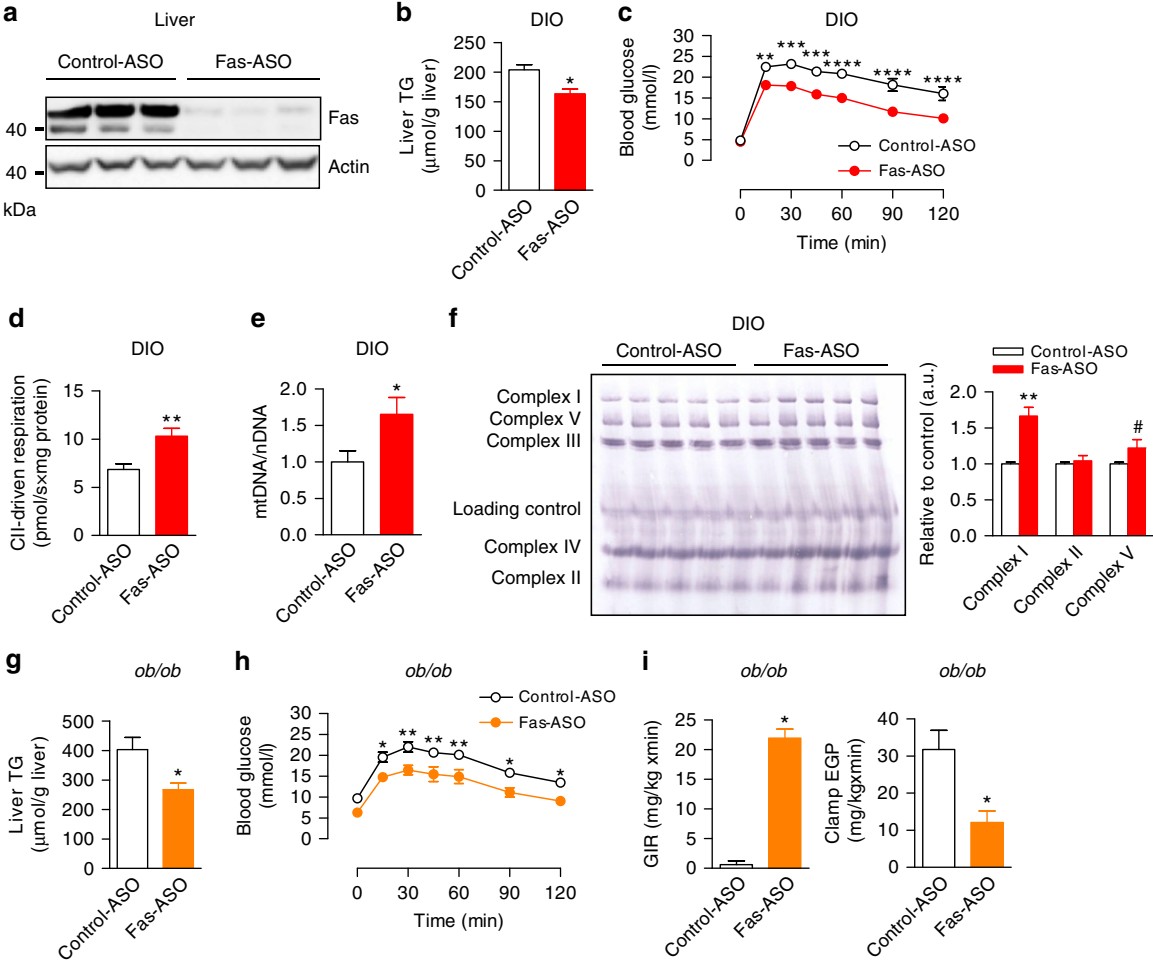

**Fig. 5** Pharmacological depletion of Fas in obese mice ameliorates metabolic dysregulation and mitochondrial function. C57BL/6J mice on HFD for 20 weeks received either antisense oligonucleotides against Fas (Fas-ASO) or control-ASO once per week (50 mg/kg body weight) during the last 10 weeks of HFD. **a** Fas protein levels in total liver lysate of mice treated with control-ASO or Fas-ASO. **b** Liver TG levels (control-ASO, $n = 5$; Fas ASO, $n = 4$) and **c** intraperitoneal glucose tolerance test (2 g/kg body weight glucose; $n = 5$) in HFD-fed mice are shown. **d** Respirometry analysis of complex II (CII)-driven respiration in liver tissue of HFD-fed control-ASO and Fas-ASO mice (control-ASO $n = 15$, Fas ASO $n = 16$). **e** Mitochondrial DNA abundance in liver ($n = 6$) and **f** blue native (BN) polyacrylamide gel electrophoresis (PAGE) of mitochondria isolated from liver ($n = 5$) of HFD-fed control-ASO and Fas-ASO mice are presented. **f–h** Leptin-deficient *ob/ob* mice were injected with Fas-ASO or control-ASO (50 mg/kg body weight) twice per week for 4 weeks. **g** Liver TG levels ($n = 6$), **h** intraperitoneal glucose tolerance test (1 g/kg body weight glucose) ($n = 6$), **i** glucose infusion rate (GIR), and endogenous glucose production (EGP) during hyperinsulinemic-euglycemic clamps are depicted (control-ASO, $n = 4$; Fas ASO, $n = 5$). Values are expressed as mean±s.e.m.; *$p < 0.05$, **$p < 0.01$, ***$p < 0.001$, ****$p < 0.0001$, and #$p = 0.09$. Statistical tests used: *t*-test for (**b**, **d–g**, **i** (*right panel*)); Mann–Whitney for (**i** (*left panel*)); and ANOVA for (**c**, **h**). DIO diet-induced obesity

and insulin resistance. However, since Fas is a prototypical death-inducing receptor and also a potential tumor suppressor that is downregulated during cancer progression[36, 37], the usefulness of such Fas-ASO treatment may be limited. Of note, liver morphology, fibrosis ($4.7 \pm 0.6\%$ Sirius red-positive area in Fas$^{F/F}$ vs. $4.9 \pm 0.6\%$ Sirius red-positive area in Fas$^{\Delta hep}$ mice, $p = 0.8$), as well as triglyceride levels ($67 \pm 5$ μmol/g liver in Fas$^{F/F}$ vs. $73 \pm 2$ μmol/g liver in Fas$^{\Delta hep}$ mice, $p = 0.2$) were not altered in 18-month-old Fas$^{\Delta hep}$ compared to Fas$^{F/F}$ mice. Additionally, loss of Fas was recently reported to reduce incidences of ovarian and liver cancer as well as tumor size in mice[38]. Consistently, Fas may promote the progression of fatty liver to cirrhosis, which is associated with liver cancer[8]. Thus, Fas activation may have both tumor-promoting and tumor-protective function. Clearly, further studies are required to explore the potential role for Fas-ASO or inhibitors of Fas signaling in the treatment of NAFLD and (hepatic) insulin resistance as well as the safety of such an approach. Of note, HFD-fed $FasL^{\Delta s/\Delta s}$ mice[22] revealed reduced liver triglyceride levels. Consequently, Fas-mediated lipid accumulation in the liver may be induced by soluble FasL, which pertinently was found to be upregulated in fatty liver disease[8]. Interestingly, genetic deletion of TRAIL (TNF-related apoptosis-inducing ligand) receptor in mice reduced HFD-induced hepatic steatosis and insulin resistance[39], suggesting that other death receptors may affect hepatic metabolism in a similar way.

In conclusion, our results reveal an important and unique role for Fas in regulating hepatic mitochondrial function and lipid metabolism. Pharmaceutical depletion of Fas or its signaling pathway may therefore emerge as a promising new avenue in the treatment of hepatic steatosis and insulin resistance.

## Methods

**Animals**. C57BL/6J mice with exon 9 of Fas flanked with LoxP sites were produced as previously described[40] and crossed with mice harboring the Cre-recombinase transgene under the control of the albumin promoter (C57BL/6J-Tg(Alb-Cre) mice) to generate hepatocyte-specific depletions (Fas$^{\Delta hep}$). Littermate mice with floxed Fas but absent Cre-recombinase (Cre) expression were used as controls (Fas$^{F/F}$). ASM KO mice on a C57BL/6N background[41] and BID KO mice on a C57BL/6J background[18] were bred in our own facility. $FasL^{\Delta s/\Delta s}$ mice (C57BL/6J background) were generated as previously described[22] and ob/ob (C57BL/6OlaHsd-Lep<ob>) mice were purchased from Harlan.

Male mice were used and housed in a specific pathogen-free environment in a 12 h light/dark cycle and fed ad libitum with regular chow diet (ProvimiKliba, Kaiseraugst, Switzerland) or high-fat diet (58 kcal% fat w/sucrose Surwit Diet, D12331, Research Diets, New Brunswick, NJ, USA). Experiments were performed at the age of 10–12 weeks (Fas$^{\Delta hep}$ and Ad-Fas mice), 26 weeks (ASO-treated mice), or 8 weeks (ob/ob mice). For all animal experiments, the sample size required to achieve adequate power was estimated on the basis of pilot work or previous experiments. Mice that did not gain weight in diet-induced obesity experiments were excluded. Where appropriate, experiments were performed in weight- and sex-matched animals. All protocols conformed to the Swiss animal protection laws and were issued by the Cantonal Veterinary Office in Zurich to the University Children's Hospital Zurich, Switzerland. All experiments with $FasL^{\Delta s/\Delta s}$ mice and control C57BL/6 mice were conducted at the Walter and Eliza Hall Institute of Medical Research and were approved by the guidelines of the Walter and Eliza Hall Institute of Medical Research Animals Ethics Committee, Victoria, Australia.

**Genotyping of animals**. All mice were genotyped by PCR with primers amplifying the Cre transgene (generating 310 bp Cre allele products and 510 bp control) and Fas (generating 319 bp WT and 399 bp "floxed" allele products).

**Recombinant adenoviruses**. Ad5-based recombinant E1- and E3-deleted adenovirus vectors expressing mouse Fas (Ad-Fas) and β-galactosidase (Ad-lacZ) were constructed and CsCl purified[42]. Viral titers were determined by plaque assay. Both transgenes are controlled by the cytomegalovirus (CMV) promoter, and the CMV promoter of Ad-Fas was modified to contain an upstream tet-repressor binding site. Ad-Fas was generated and amplified in 911 helper cells expressing the tet-repressor. Viruses were injected into the tail vein of 8–10-week-old C57BL/6J mice at a dose of $1 \times 10^9$ plaque-forming units.

**Mouse Fas ASO**. ASOs were synthesized using an Applied Biosystems 380B automated DNA synthesizer (Perkin Elmer–Applied Biosystems, Foster City, CA, USA) and purified by ion-exchange high-performance liquid chromatography (HPLC) using a linear gradient of buffers A and B. Buffer A: 50 mM NaHCO$_3$, buffer B: 1.5 M NaBr, 50 mM NaHCO$_3$—both buffers in acetonitrile/water 3:7 (v:v). Purified ASOs were desalted using C18 reverse-phase cartridges and analyzed by HPLC. The ASO identification numbers and sequences are as follows: mismatch control oligonucleotide (control-ASO), ISIS 141923, 5′-CCTTCCCTGAAGGTT CCTCC-3′; Fas-ASO, ISIS22023, 5′-TCCAGCACTTTCTTTTCCGG-3′, BID-ASO, ISIS 119935, 5′-GACCATGTCCTGGCCAGAAA-3′. The backbone was comprised of phosphorothioate and the first and last five bases of the ASOs have a 2′-O-methoxyethyl modification. ASOs were resuspended and delivered in 0.9% saline. HFD-fed C57BL/6J mice received 50 mg ASO/kg body weight once per week for 10 weeks or 4 consecutive days as described[21]; leptin-deficient ob/ob (C57BL/6OlaHSD-Lep<ob>) mice were treated with 50 mg/kg body weight twice a week for 4 weeks. Mice with liver-specific Fas overexpression received 3 injections at 50 mg ASO/kg body weight per week starting 1 week before adenovirus injection.

**Liver triglyceride measurement**. Liver triglycerides were measured from 50 mg of liver tissue according to the method of Bligh and Dyer[43] and quantified with an enzymatic assay (Roche Diagnostics, Rotkreuz, Switzerland). Prior to tissue collection, mice were fasted for 5 h except for liver-specific Fas-knockout mice, which were fasted overnight.

**Liver triglyceride secretion**. Ad-Fas mice were fasted for 6 h and injected intravenously with the lipase inhibitor tyloxapol (500 mg/kg; Sigma) prior to blood collection at 0, 1, 2, 4, 6, and 24 h after injection. The collected blood samples were used for TG measurements and the very low density lipoprotein–TG production rate was calculated from the slope of the plasma TG vs. time curve.

**Glucose and pyruvate tolerance test**. Glucose (1 g/kg body weight for ob/ob mice; 2 g/kg body weight for C57BL/6J) and pyruvate (2 g/kg body weight) were injected intraperitoneally in overnight fasted mice and in mice fasted for 3 h, respectively. Blood glucose concentration was measured in blood from tail-tip bleedings using a glucometer (Accu-Check Aviva, Roche Diagnostics).

**Hyperinsulinemic-euglycemic clamp studies**. Hyperinsulinemic-euglycemic clamp studies were performed in freely moving mice as described[44]. Insulin was infused at a constant rate (18 mU/kg×min) and glucose infusion rate was calculated once glucose infusion reached a more or less constant rate with blood glucose levels at 5 mmol/l (~80–100 min after the start of insulin infusion). Thereafter, blood glucose was kept constant at 5 mmol/l for 20 min and glucose infusion rate was calculated. The glucose disposal rate was calculated by dividing the rate of [3-$^3$H] glucose infusion by the plasma [3-$^3$H]glucose-specific activity[45, 46]. Endogenous glucose production during the clamp was calculated by subtracting the glucose infusion rate from the glucose disposal rate[45, 46]. Insulin-stimulated glucose disposal rate was calculated by subtracting basal endogenous glucose production (equal to basal glucose disposal rate) from glucose disposal rate during the clamp[47].

**Ceramide and diacylglycerol analysis in liver tissue**. Lipids were extracted from ~30 mg frozen liver tissue according to the method of Bligh and Dyer[43], with 50 mM potassium phosphate buffer in the aqueous layer and spiked with appropriate internal standards. Extracted samples were evaporated to dryness and redissolved in methanol. The extracts were diluted 5 times in methanol/water (4:1 v/v) with 10 mM ammonium acetate prior to liquid chromatography/tandem mass spectrometry analysis to enhance chromatography and to induce ionization. Chromatographic separation was performed on a nanoAcquity UPLC system equipped with a HSS T3 column (Waters, Milford, MA, USA). The chromatographic conditions were adopted from Castro-Perez et al.[48]. A Waters Synapt G2 HDMS mass spectrometer was employed for the analysis of the lipid composition. Data acquisition and processing were performed with the MassLynx software (Waters), and the targeted lipids were identified based on their exact mass using the LipidBlast library and the NIST MSsearch software[49].

**Histology**. Liver tissues were fixed in 4% buffered formalin and embedded in paraffin. Sections were cut and stained with hematoxylin and eosin.

**Fatty acid oxidation**. A piece of 100–200 mg liver tissue was homogenized in sodium chloride-Tris-EDTA buffer, centrifuged at $420 \times g$ for 10 min at 4 °C, and the supernatant was incubated for 40 min at 37 °C in a reaction mixture containing $^{14}$C-radiolabeled oleic acid, 2 mM ATP, 0.5 mM dithiothreitol, and measured for acid-soluble metabolites and trapped $CO_2$.

**Oxygen consumption rate**. Liver tissue was homogenized in mir05 medium (EGTA 0.5 mM, MgCl$_2$.6H$_2$O 3 mM, K-lactobionate 60 mM, Taurine 20 mM, KH$_2$PO$_4$ 10 mM, HEPES 20 mM, Sucrose 110 mM, BSA, essentially fatty acid free 1 g/l). Oxygen consumption rate was then assessed using high-resolution

respirometry (Oroboros Oxygraph-2k, Oroboros Instruments, Innsbruck, Austria). Succinate (9.6 mM), ADP (4.8 mM), and rotenone (0.1 mM) were sequentially added to stimulate complex II-driven oxygen consumption[50].

**Mitochondrial isolation.** Liver tissue was homogenized in mitochondria isolation buffer (IBc) that contains 10 ml of 0.1 M Tris-MOPS, 1 ml of EGTA/Tris and 20 ml of 1 M sucrose and 69 ml of distilled water, pH 7.4. Homogenates were centrifuged at 600×g for 10 min at 4 °C to discard the debris leaving mitochondria in the supernatant. Supernatant was spun down at 7,000×g for 10 min at 4 °C to pellet the mitochondria. The pellets were washed once with 1 ml IBc. The mitochondria pellet was used for blue native PAGE.

**BN-PAGE and in-gel activity.** A high-resolution BN-PAGE was performed to analyze mitochondrial complexes expression and activity using the Native-PAGE Novex Bis-Tris Gel system (Invitrogen, Buchs, Switzerland) according to the manufacturer's instructions. Mitochondria were isolated from liver tissue and resuspended in Native-PAGE sample buffer containing 0.5% *n*-dodecyl-β-D-maltoside (Invitrogen) for mitochondrial complexes' expression or digitonin (Thermo Fisher Scientific, Reinach, Switzerland) 4 g/g protein for activity. Equal amounts of mitochondrial proteins as determined by Bradford protein assay were loaded on a Native-PAGE Novex 3–12% gradient Bis-Tris acrylamide gel (Invitrogen). Gels were transferred onto polyvinylidene difluoride membranes using iBlot gel transfer system (Thermo Fisher Scientific) and immunoblotted with OxPhos complex antibody cocktail (Mitosciences, Eugene, OR, USA) or incubated with substrate solutions to assess in-gel activity. Complex IV substrate solution: 50 mg diaminobenzidine, 100 mg cytochrome *c*, and 90 ml 50 mM phosphate buffer pH 7.4 to 10 ml with double-distilled water (ddH₂O). Same gel was washed with water and then incubated with complex I substrate solution: 40 µl 1 M Tris-HCl, pH 7.4, 2 mg NADH, 50 mg nitrotetrazolium blue chloride (NTB) to 20 ml with ddH₂O. Another gradient gel was used to assess complex II activity: 200 µl 1 M sodium succinate, 25 mg NTB, 8 µl 250 mM phenazinemethosulfate (from 250 mM stock in dimethyl sulfoxide), 50 µl 1 M Tris-HCl to 10 ml with ddH₂O. Reactions were stopped by washing with 10% acetic acid for 5 min followed by water for 5 min. Gels were scanned at different time points of incubation to follow kinetics of the signal before saturation.

**Acid sphingomyelinase activity.** Samples were shock frozen and lysed in 250 mM sodium acetate (pH 5.0) and 1% NP40 for 15 min on ice. The samples were further homogenized by two rounds of sonication for 10 s each using a tip sonicator. Aliquots of the lysates were diluted to 250 mM sodium acetate (pH 5.0), 0.1% NP40 and incubated with 50 nCi per sample [¹⁴C]sphingomyelin (Perkin Elmer, Waltham, MA, USA; 52 mCi/mmol) for 30 min at 37 °C. The substrate was dried prior to the assay, resuspended in 250 mM sodium acetate (pH 5.0), 0.1% NP40, and bath-sonicated for 10 min to promote the formation of micelles. The reaction was stopped by the addition of 800 µL chloroform/methanol (2:1, v/v), phases were separated by centrifugation, and an aliquot of the upper, aqueous phase was measured by using liquid scintillation counting to determine the release of [¹⁴C] phosphorylcholine from [¹⁴C]sphingomyelin as a measure of acid sphingomyelinase activity.

**Determination of plasma parameters.** Plasma insulin and FFA levels were determined as previously described[51]. Plasma leptin and cytokine levels were determined with mouse Procarta Cytokine Assay Kit (Affymetrix). Plasma adiponectin levels were measured with an ELISA kit (Axxora, San Diego, CA, USA). Triglyceride concentrations were determined using a colorimetric assay (Sigma, St Louis, MO, USA). ALT/AST levels were measured using a multiple biochemical analyzer (Dri-Chem 4000i; Fujifilm, Tokyo, Japan).

**RNA extraction and quantitative reverse transcription-PCR.** Total RNA was extracted and reverse transcribed[52]. The following primers were used: Acox1, Mm00443579_m1; CPT1α, Mm00550438_m1; ACC1 Mm01304289_m1; FAS Mm00662319_m1; SCD-1 Mm01197142_m1; SREBP1c Mm00550338_m1; CD36 Mm00432403_m1; MTTP Mm00435015_m1; PPARα, Mm00627559_m1; KC, Mm00433859_m1; MCP-1 Mm00441242_m1; TNFα, Mm00443258_m1; CD68, Mm03047343_m1 (Applied Biosystems, Foster City, CA, USA).

**Insulin signaling.** After an overnight fast, mice were injected 5 units normal human insulin (Actrapid, Novo Nordisk, Denmark) into the portal vein. Liver samples were collected 2 min after injection and proteins were extracted from tissues for western blot analysis.

**Western blotting.** Tissue samples were homogenized in a buffer containing 150 mM NaCl, 50 mM Tris-HCl (pH 7.5), 1 mM EGTA, 1% NP40, 0.25% sodium deoxycholate, 1 mM sodium vanadate, 1 mM NaF, 10 mM sodium β-glycerophosphate, 100 nM okadaic acid, 0.2 mM phenylmethylsulfonyl fluoride, and a 1:1,000 dilution protease inhibitor cocktail (Sigma). Protein concentration was determined using bicinchoninic acid assay and equivalent amounts of protein were resolved by LDS-(lithium dodecyl sulfate)-PAGE (4–12% gel; NuPAGE, Invitrogen). Proteins were transferred to a nitrocellulose membrane (0.2 µm, Bio-Rad) and blocked for 1

h in 5% non-fat dry milk (Bio-Rad) resolved in Tris-buffered saline, containing 1% Tween-20. Membranes were incubated overnight at 4 °C on a rocking platform with respective primary antibodies. The following primary antibodies were used: anti-Fas (05–351; clone 7C10), anti-actin (MAB1501) (Millipore, Billerica, MA, USA), anti-caspase 8 (ALX-804-448; 3B10) (Enzo Life Sciences, Farmingdale, NY, USA), anti-cleaved caspase 3 (9661), anti-PARP (9542S), anti-cytochrome *c* (4272), anti-HSP90 (4877), anti-Akt (9272) and anti-phospho-Akt (Thr308; 9275S) (Cell Signaling, Danvers, MA, USA), anti-cleaved-Bid (ab10640) (Abcam, Cambridge, UK), anti-Grp78 (sc-1051) (Santa Cruz Biotechnology, Dallas, TX, USA), anti-SREBP1 (557036) (BD Biosciences, Allschwil, Switzerland), and anti-BID[53].

**Metabolic cage analysis.** Locomotion, food and water intake, O₂ consumption, and CO₂ production were determined for single-housed mice during a 24 h period in a metabolic and behavioral monitoring system (PhenoMaster, TSE Systems, Bad Homburg, Germany).

**Data analysis.** Data are presented as means±s.e.m. Data distribution was assessed via Shapiro–Wilk test using SPSS 13.0 (Chicago, IL, USA). Normally distributed data with similar variance between the groups were analyzed by unpaired two-tailed Student's *t*-test, one-way analysis of variance (ANOVA) with Tukey or Newman–Keuls correction for multiple group comparisons or two-way ANOVA with Bonferroni multiple comparisons. For nonparametric data the Mann–Whitney test was used. All statistical tests were calculated using the GraphPad Prism 5.04 (GraphPad Software, San Diego, CA, USA). P-values of <0.05 were considered to be statistically significant. Power calculation analysis was not performed. The evaluator was blinded to the identity of a specific sample as far as the nature of the experiment allowed it.

**Data availability.** The data supporting the findings of this study are available within the article, its Supplementary Information files, or from the corresponding author on reasonable request.

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

## Acknowledgements

This work was supported by grants from the Swiss National Science Foundation (310030-141238 as well as 310030-160129 to D.K. and 31003A-140780 to J.A.), the Ecole Polytechnique Fédérale de Lausanne, the Krebsforschung Schweiz/Swiss Cancer League (KFS-3082-02-2013 to J.A.), the Foundation for Research at the Medical Faculty, University of Zurich, Switzerland (to F.I.), a Cancer Australia and Cancer Council New South Wales project grant 1047672 (to L.A.O'R.), and an NHMRC (Australia) infrastructure grant, Independent Research Institutes Infrastructure Support Scheme Grant 361646, the Victorian State Government, Australia (OIS grant to L.A.O'R.). J.A. is the Nestlé Chair in Energy Metabolism. E.P. was supported by the Academy of Finland. We greatly acknowledge Professor Eugen Schoenle (University Children's Hospital Zurich, Switzerland) and Giatgen Spinas (University Hospital Zurich, Switzerland) for continuous support, Professor Mathias Heikenwälder/Professor Adriano Aguzzi (University Hospital Zurich, Switzerland) for providing Alb-Cre expressing mice, Professor Alexander Chervonsky (University of Chicago, USA) for providing Fas-lox mice, Mirzet Delic, Thea Fleischmann (University Hospital Zurich, Switzerland), Manuel Klug (Laboratory Animal Service Center (LASC)), and Ann Lin (The Walter and Eliza Hall Institute of Medical Research, Australia) for great technical support, and Endre Laczko from the Functional Genomics Center Zurich (FGCZ) for assistance regarding diacylglycerol and ceramide determination.

## Author contributions

F.I. designed and performed experiments, analyzed data, and wrote the manuscript. S.W. designed and performed experiments. V.L., S.S., F.C.L., R.D., M.C.F., T.D.C., L.A.O'R., and E.P. performed experiments. Y.K. provided Fas- and BID-antisense oligonucleotides. S.H. provided Fas and control adenoviruses. E.G. analyzed ASM activity and provided ASM-knockout mice. A.G. provided BID-knockout mice. M.S. and J.A. gave conceptual advice and supervised the experiments. E.G. and A.G. gave conceptual advice. D.K. designed experiments, analyzed data, and wrote the manuscript. All authors reviewed and commented on the manuscript.

## Additional information

**Competing interests:** The authors declare no competing financial interests.

