## [Peer Review File · Nature Communications]

Editorial Note: *This manuscript has been previously reviewed at another journal that is not operating a transparent peer review scheme. This document only contains reviewer comments and rebuttal letters for versions considered at Nature Communications. Mentions of prior referee reports have been redacted.*

Reviewers' comments:

Reviewer #1 (Remarks to the Author):

The manuscript (NCOMMS-16-16400A) entitled, "Fas (CD98) controls hepatic lipid metabolism by regulating mitochondrial function" is an original manuscript implicating Fas signaling in hepatic steatosis. The manuscript has been revised from a previous submission to Nature Medicine. There are two issues which need to be addressed.

MAJOR COMMENTS:

1. The authors have not examined the effect of Fas signaling on the master regulators of hepatic steatosis SREBP 1c for fatty acids and lipids. Assessing the effect of Fas signaling on lipogenic regulators is quite important either as a mechanism for decreased hepatic steatosis in response to FAS or as a critical control.

2. The authors do not identify caspase 3 activation as a result of Fas overexpression. It is difficult to reconcile how Fas signaling in the absence of death receptor mediated cell death would influence hepatic steatosis. The hepatic steatosis is Bid-dependent thereby implicating its cleavage by pro-apoptotic Fas signaling; generation of truncated Bid or tBid should result in mitochondrial release of cytochromes C and the activation of caspase 3. It would be important to determine whether the degree of Bid cleavage or tBid generation is sufficient to release cytochrome C as was requested in the first review of this manuscript. These experiments were not performed and still need to be executed. Further information on how tBid would result in mitochondrial dysfunction without causing cell death needs to be clarified in the current study. The authors need to clarify this area of Fas biology in their models. For example, is XIAP expression increased under the conditions of these experiments and would it be expected to inhibit caspase 3 induced cell death. The authors could perform immunohistochemistry for SMAC or cytochrome C release from mitochondria. These experiments are critical to interpret these paradigms. Without further clarification of the relationship between mitochondrial dysfunction and the cell death pathways the experiments in the current form do not fully support the conclusions of the manuscript.

Reviewer #3 (Remarks to the Author):

The authors have done an excellent job in addressing the original concerns. I have a few remaining issues.

Major:

Original comment #1. The assessment of fatty acid metabolism is unsatisfactory and a major limitation of the present work. Contrary to the authors claims, these experiments are not beyond the scope of the manuscript and are, in fact, central to understanding the effect of Fas on steatosis development. Assessing mRNA expression at a single timepoint does not inform on metabolism.

(Previous comment: Lipid metabolism. Figure 3b-c. The fatty acid metabolism experiments are incomplete and do not take into consideration the complexity of hepatic lipid metabolism. The authors should show the rates of fatty acid uptake (i.e. is decreased uptake mediating the decreased oxidation rate) and also the rates of fatty acid conversion into other lipids (notably triglycerides given that steatosis is a primary outcome measure). In fact, in the experiments described, the rate of fatty acid oxidation is typically significantly less than fatty acid storage into complex lipids, so quantitatively, the oxidation pathway may be relatively unimportant. It is also unsatisfactory to state that lipogenic gene expression was unchanged (data not shown) and therefore that altered lipogenesis is not contributing to the steatosis in these mice (Ad-Fas). This needs to be assessed directly. Finally, defects in TG secretion should be assessed in mice, as this impacts lipid balance in the liver).

Original comment #4. The authors have provided some information to address issues relating to tissue damage, fibrosis and inflammation. This data should be included in a revised manuscript.

(Previous comment: The authors address the impact of Fas on liver metabolism, but as they are aware, NALFD progresses with time to cause inflammation and fibrosis. I think the major piece of missing data is the examination of liver inflammation, liver function and liver histology in longer-term models of NAFLD. Is the decrease in TG levels (which are not trivial but not astronomical with Fas ablation) sufficient to prevent the progression of disease? This information would be essential when assessing the veracity of their conclusions and for pursuing therapeutic targeting of this pathway.)

Minor comment #2. Include all the requested data for the insulin clamp including the insulin infusion rate, the endogenous Ra and insulin levels during the clamp. This is important information.

Minor comment #4. Figure S6. Demonstration of Fas overexpression in the adenovirus studies should be included in the main text. Same for Figure S15, which should be moved to Figure 4 (i.e. ASO expts). This data is essential to show to demonstrate that your experiments have worked!

Point-by-point response to Reviewers' comments NCOMMS-16-16400-T

Answers to Reviewer #1:

We thank the reviewer for his/her insightful and important comments.

1. The authors have not examined the effect of Fas signaling on the master regulators of hepatic steatosis SREBP 1c for fatty acids and lipids. Assessing the effect of Fas signaling on lipogenic regulators is quite important either as a mechanism for decreased hepatic steatosis in response to FAS or as a critical control.

We have now analysed protein levels of cleaved SREBP1 in HFD-fed control (Fas^{F/F}) and liver-specific Fas knockout (Fas^{Δhep}) mice and found similar levels between the groups (Supplementary Fig. 3h of revised manuscript). Consistently, mRNA expression levels of SREBP1c and its downstream targets ACC1, FAS and SCD1 were unchanged between the two genotypes (Supplementary Fig. 3i of revised manuscript). Hence, SREBP1-induced gene expression is not affected in Fas^{Δhep} mice.

Moreover, we analysed cleaved SREBP1 in mice with hepatic overexpression of Fas. As shown in Supplementary Fig. 3e of the revised manuscript, cleaved SREBP1 was slightly lower in Ad-Fas-injected mice. In agreement, mRNA expression of SREBP1c and downstream targets such as ACC1, FAS and SCD1 was reduced by 40-50% (Supplementary Fig. 3f of revised manuscript). Such data suggest that the lipogenic SREBP1-pathway is not induced in Fas-overexpressing mice.

2. The authors do not identify caspase 3 activation as a result of Fas overexpression. It is difficult to reconcile how Fas signaling in the absence of death receptor mediated cell death would influence hepatic steatosis. The hepatic steatosis is Bid-dependent thereby implicating its cleavage by pro-apoptotic Fas signaling; generation of truncated Bid or tBid should result in mitochondrial release of cytochromes C and the activation of caspase 3. It would be important to determine whether the degree of Bid cleavage or tBid generation is sufficient to release cytochrome C as was requested in the first review of this manuscript. These experiments were not performed and still need to be executed. Further information on how tBid would result in mitochondrial dysfunction without causing cell death needs to be clarified in the current study. The authors need to clarify this area of Fas biology in their models. For example, is XIAP expression increased under the conditions of these experiments and would it be expected to inhibit caspase 3 induced cell death. The authors could perform immunohistochemistry for SMAC or cytochrome C release from mitochondria. These experiments are critical to interpret these paradigms. Without further clarification of the relationship between mitochondrial dysfunction and the cell death pathways the experiments in the current form do not fully support the conclusions of the manuscript.

We agree with the Reviewer that the lack of caspase 3 activation in Fas-overexpressing mice is somewhat surprising (Supplementary Fig. 3c of revised manuscript). As suggested, we analysed XIAP protein levels and found them to be elevated in Ad-Fas compared to Ad-LacZ mice (see below; n=6-7; Co: total liver lysate of a C57BL/6J mouse on HFD for 20 weeks).

Moreover, we also analysed mitochondrial cytochrome c release in Ad-lacZ and Ad-Fas mice by Western blot. As shown in Supplementary Fig. 3b of the revised manuscript, no cytochrome c was detectable in the cytosolic fraction of livers isolated from Ad-Fas mice indicating that the degree of Fas-mediated BID cleavage (Fig. 4b of revised manuscript) may not be sufficient to trigger cytochrome c release and subsequent cleavage of caspase 3. Alternatively, increased protein levels of X-linked inhibitor of apoptosis protein (XIAP) (see above) may explain lack of caspase 3 cleavage in Fas-overexpressing mice since XIAP can prevent cytochrome c-induced caspase 3 cleavage (Deveraux QL et al. *EMBO J* 1998; 17(8): 2215-23). Hence, while our experiments in BID KO as well as BID ASO-treated mice strongly suggest an involvement of BID in Fas-mediated hepatic lipid accumulation, further studies are needed to shed more light on the complex molecular mechanisms involved in the activation of BID-induced apoptotic signaling (Kantari C and Walczak H. *Biochim Biophys Acta* 2011; 1813: 558-563).

In contrast to our *in vivo* findings, treatment of primary hepatocytes with 2ng/ml FasL induced cytochrome c release as well as cleavage of caspase 3 (see below).

These experiments suggest that treatment of primary hepatocytes with 2ng/ml FasL induces apoptosis, making any interpretation of data collected in this model rather difficult. Consequently, we decided to remove the few experiments performed in primary hepatocytes from the revised manuscript (Fig. 3a, b and Fig. 4a, f of the previously submitted manuscript) and to rely solely on *in vivo* data, where we found no signs of altered apoptosis (see above). Nevertheless, we cannot exclude that undetected apoptosis may have contributed to the observed phenotype in Fas-overexpressing mice. Such statement was now added to the *Discussion* section of the revised manuscript.

Answers to Reviewer #3:

We thank the reviewer for his/her constructive comments and suggestions and we are delighted to learn that he/she found that we have done an excellent job in addressing the original concerns

Major:

Original comment #1. The assessment of fatty acid metabolism is unsatisfactory and a major limitation of the present work. Contrary to the authors claims, these experiments are not beyond the scope of the manuscript and are, in fact, central to understanding the effect of Fas on steatosis development. Assessing mRNA expression at a single timepoint does not inform on metabolism.

(Previous comment: Lipid metabolism. Figure 3b-c. The fatty acid metabolism experiments are incomplete and do not take into consideration the complexity of hepatic lipid metabolism. The authors should show the rates of fatty acid uptake (i.e. is decreased uptake mediating the decreased oxidation rate) and also the rates of fatty acid conversion into other lipids (notably triglycerides given that steatosis is a primary outcome measure). In fact, in the experiments described, the rate of fatty acid oxidation is typically significantly less than fatty acid storage into complex lipids, so quantitatively, the oxidation pathway may be relatively unimportant. It is also unsatisfactory to state that lipogenic gene expression was unchanged (data not shown) and therefore that altered lipogenesis is not contributing to the steatosis in these mice (Ad-Fas). This needs to be assessed directly. Finally, defects in TG secretion should be assessed in mice, as this impacts lipid balance in the liver).

As suggested, we now analysed hepatic triglyceride secretion in Ad-lacZ and Ad-Fas mice after Triton-induced hypertriglyceridemia. As shown in Supplementary Fig. 3g of the revised manuscript, triglyceride secretion was decreased in Fas-overexpressing mice. In line, MTP mRNA expression was reduced by 30% in these mice (Supplementary Fig. 3f). Such data indicate that reduced triglyceride secretion may contribute to elevated liver steatosis observed in Ad-Fas mice. A statement indicating such, has been added to the *Discussion* section of the revised manuscript. In addition, SREBP1 and its downstream targets were analysed in HFD-fed Fas^{Δhep} and Fas-overexpressing mice and these do not seem to be significantly modulated by Fas (see answer to comment 1 of Reviewer 1).

As suggested, we measured fatty acid uptake into isolated primary hepatocytes. As shown below, treatment of primary hepatocytes with 2ng/ml FasL did not significantly affect palmitate uptake.

In addition, MTP mRNA expression in primary hepatocytes and ApoB concentration in the supernatant were determined to investigate the potential impact of FasL-treatment on triglyceride secretion. In line with findings in Fas-overexpressing mice, treatment of primary hepatocytes with 2ng/ml FasL blunted ApoB concentration in the supernatant and reduced MTP mRNA expression by ~35% (see below) indicating reduced triglyceride secretion upon Fas activation.

Of note, treatment of hepatocytes with a lower dose of FasL (1ng/ml) did neither affect MTTP mRNA expression nor expression of genes involved in beta oxidation (see below).

Taken together, treatment of hepatocytes with 2 ng/ml FasL decreased beta oxidation (Fig. 3b of previously submitted manuscript), decreased TG secretion but had no impact on fatty acid uptake. However, treatment of primary hepatocytes with 2 ng/ml FasL induced apoptosis (see answer to comment 2 of Reviewer 1), making interpretation of obtained data difficult. Consequently, we decided to remove the few experiments performed in primary hepatocytes from the revised manuscript (Fig. 3a, b and Fig. 4a, f of previously submitted manuscript) and to fully concentrate on the *in vivo* data.

Original comment #4. The authors have provided some information to address issues relating to tissue damage, fibrosis and inflammation. This data should be included in a revised manuscript.

(Previous comment: The authors address the impact of Fas on liver metabolism, but as they are aware, NALFD progresses with time to cause inflammation and fibrosis. I think the major piece of missing data is the examination of liver inflammation, liver function and liver histology in longer-term models of NAFLD. Is the decrease in TG levels (which are not trivial but not astronomical with Fas ablation) sufficient to prevent the progression of disease? This information would be essential when assessing the veracity of their conclusions and for pursuing therapeutic targeting of this pathway.)

As previously mentioned, we can provide data obtained from 18 months-old chow-fed animals. As mentioned in the previously submitted manuscript, liver morphology was not altered in these mice suggesting that hepatic Fas depletion had no tumor promoting effect in our mouse model. We have now added a statement regarding similar degree of triglyceride levels and fibrosis in 18 months old chow fed Fas^{F/F} and Fas^{Δhep} mice to the *Discussion* of the revised manuscript.

Minor comment #2. Include all the requested data for the insulin clamp including the insulin infusion rate, the endogenous Ra and insulin levels during the clamp. This is important information.

Glucose infusion rates as well as rates of endogenous Ra (glucose turnover) are now included in the revised manuscript (Fig. 1f, 2d, 5i; Supplementary Fig. 1i,k; 2c, d; 5b, c). Unfortunately, we do not have plasma samples left to determine insulin levels during the clamps.

Minor comment #4. Figure S6. Demonstration of Fas overexpression in the adenovirus studies should be included in the main text. Same for Figure S15, which should be moved to Figure 4 (i.e. ASO expts). This data is essential to show to demonstrate that your experiments have worked!

We now moved Western blots confirming Fas overexpression and ASO-induced Fas depletion to the main Figures (Fig. 2a, 5a).

REVIEWERS' COMMENTS:

Reviewer #1 (Remarks to the Author):

The authors have satisfactorily addressed my prior comments.

Reviewer #2 (Remarks to the Author):

The authors have adequately addressed my queries through the addition (and removal) of important data relating to hepatic triglyceride metabolism and some data on liver fibrosis. I have no further comments.